# Relationship between Gut, Blood, Aneurysm Wall and Thrombus Microbiome in Abdominal Aortic Aneurysm Patients

**DOI:** 10.3390/ijms25168844

**Published:** 2024-08-14

**Authors:** Éva Nemes-Nikodém, Gergő Péter Gyurok, Zsuzsanna A. Dunai, Nóra Makra, Bálint Hofmeister, Dóra Szabó, Péter Sótonyi, László Hidi, Ágnes Szappanos, Gergely Kovács, Eszter Ostorházi

**Affiliations:** 1Department of Medical Microbiology, Semmelweis University, 1089 Budapest, Hungary; nemes-nikodem.eva@med.semmelweis-univ.hu (É.N.-N.); makra.nora@med.semmelweis-univ.hu (N.M.); hofmeister.balint@semmelweis.hu (B.H.); szabo.dora@semmelweis.hu (D.S.); 2Department of Vascular and Endovascular Surgery, Heart and Vascular Center, Semmelweis University, 1122 Budapest, Hungary; gyurok.gergo@gmail.com (G.P.G.); sotonyi.peter1@semmelweis.hu (P.S.); hidi.laszlo@med.semmelweis-univ.hu (L.H.); drszappanos@gmail.com (Á.S.);; 3HUN-REN-SU Human Microbiota Research Group, 1052 Budapest, Hungary; zsuzsanna.dunai@gmail.com; 4Department of Rheumatology and Clinical Immunology, Semmelweis University, 1023 Budapest, Hungary; 5Department of Dermatology, Venereology and Dermatooncology, Semmelweis University, 1085 Budapest, Hungary

**Keywords:** vessel wall microbiome, aneurysm, thrombus microbiome, blood microbiome, stool microbiome

## Abstract

Previous research confirmed gut dysbiosis and translocation of selected intestinal bacteria into the vessel wall in abdominal aortic aneurysm patients. We studied the stool, blood, thrombus and aneurysm microbiomes of 21 abdominal aortic aneurysm patients using 16S rRNA sequencing. Our goals were to determine: 1. whether the microbiome characteristic of an aneurysm differs from that of a healthy vessel, 2. whether bacteria detectable in the aneurysm are translocated from the gut through the bloodstream, 3. whether the enzymatic activity of the aneurysm microbiome can contribute to the destruction of the vessel wall. The abundance of *Acinetobacter*, *Burkholderia*, *Escherichia*, and *Sphingobium* in the aneurysm samples was significantly higher than that in the microbiome of healthy vessels, but only a part of these bacteria can come from the intestine via the blood. Environmental bacteria due to the oral cavity or skin penetration route, such as *Acinetobacter*, *Sphingobium*, *Enhydrobacter*, *and Aquabacterium*, were present in the thrombus and aneurysm with a significantly higher abundance compared to the blood. Among the enzymes of the microbiome associated with the healthy vessel wall, Iron-chelate-transporting ATPase and Polar-amino-acid-transporting ATPase have protective effects. In addition, bacterial Peptidylprolyl isomerase activity found in the aneurysm has an aggravating effect on the formation of aneurysm.

## 1. Introduction

Abdominal aortic aneurysm (AAA) is a permanent, localized dilatation of the abdominal aorta either presenting a diameter of more than 30 mm or exceeding the normal vessel diameter by 50% [1]. Major risk factors for developing an AAA are smoking, male sex, family history, advanced age, hypertension, hypercholesterolemia, obesity and various cardiovascular/peripheral diseases [2]. The background of the pathological change in the vessel wall is immune cell-mediated inflammation and degradation of the medial layer [3]. Recently, several studies have shown that a correlation exists between the composition of the gut microbiome and the development and progression of AAA [4,5,6]. Fecal transplantation with altered microbiota of AAA patients significantly increases the rate of aneurysm formation, rupture and mortality in AAA mice, whereas healthy human microbiota induces the opposite effect [5]. Since the diet affects the composition of the gut microbiome, and thus the immunological and inflammatory processes in the body, it is worth conducting further research on the relationship between diets affecting the gut microbiome and the progression of AAA [7].

Studies investigating human samples of AAA patients or experimental animal studies have most often connected the relationship between the faecal microbiome and AAA [5,8,9,10,11,12,13,14], although the role of the oral microbiome in the pathomechanism is also scrutinized [15,16]. Gut or oral microbiota may indirectly or directly affect the inflammatory destruction of the vessel wall. Indirectly, as circulating bacterial metabolites contribute to inflammatory and apoptotic processes, or directly, as the translocated microbiota triggers increased infiltration of inflammatory cells in the blood vessel [8,9].

DNA of selected bacteria originating from the intestinal or oral microbiome is positively detectable in the vascular, thrombus or blood samples of AAA patients, suggesting that bacteria can participate in the progression of the aneurysm with their direct presence [17,18,19,20]. The DNA of the 11 enteral bacterial species targeted by Nakayama et al. could not be detected in all examined aneurysm and blood samples, and the related samples from the same patients did not always yield the same bacterial presence results [17]. In our previous study, we confirmed that bacterial DNA can also be detected in the healthy vessel wall by sequencing the bacterial 16S rRNA V3-V4 region from the artery wall samples of healthy organ donors. The composition of the microbiome detected in this way is significantly different from the composition of oral or fecal microbiome [21].

In the current study, we were looking for answers to three questions, including how the microbiome detectable in the aneurysm wall differs from the microbiome in the healthy vessel wall. Additionally, we studied whether the faecal microbiome origin and translocation through the bloodstream of the bacteria are detectable in the aneurysm wall and thrombus. Thirdly, by using the PICRUST2 v2.5.2 tool and examining the predicted enzyme activity of the detected bacteria, we attempt to establish a connection to the direct pathogenic role of the bacteria in the creation of AAA.

## 2. Results

After sequencing the 16S rRNA V3-V4 region of the aneurysm vessel wall, thrombus, blood and stool samples of 21 patients who underwent surgery at the Department of Vascular and Endovascular Surgery, Heart and Vascular Center of Semmelweis University, we obtained sufficient read numbers in each sample to be able to compare and evaluate the data. The median number of reads within one sample, regardless of the sample type was 242591 (IQR: 21678).

In response to the question of whether the aneurysm wall has a microbiome with a special composition, we compared the obtained data with the results of our previous work examining the microbiome of healthy vessel walls. Figure 1 shows that there is a significant difference between the two groups based on the Chao1 alpha diversity (Figure 1A) (*p* = 0.0001) and Bray Curtis beta diversity Principal Coordinate analysis (PCoA) (Figure 1B) (*p* = 0.001).

At the phylum level, identical four taxa are present in both vessel samples, *Proteobacteria*, *Firmicutes*, *Bacteroidetes* and *Actinobacteria*. However, there is a significant difference in the abundance of phyla between the two groups. In contrast to the 34% abundance of *Proteobacteria* in the healthy vessel wall, this phylum was present in the aneurysm wall at 71%. The occurrence of 29% *Firmicutes*, 10% *Bacteroidetes* and 15% *Actinobacteria* in the healthy vessel wall decreased to 14%, 4% and 4%, respectively, in the aneurysm wall. The most abundant genera in the healthy vessel microbiome were *Pseudomonas* (from *Proteobacteria* phylum), *Staphylococcus* (from *Firmicutes* phylum) and *Corynebacterium* (from *Actinobacteria* phylum). The genera belonging to the *Proteobacteria* phylum elevated in the aneurysm vessel wall are *Acinetobacter*, *Burkholderia*, *Escherichia*, *Escherichia-Shigella* and *Sphingobium*. From the *Firmicutes* phylum, the genus *Finegoldia* was present in higher abundance, and the genus *Staphylococcus* in lower abundance in the aneurysm wall than in the healthy vessel wall. The greater presence of *Corynebacterium* and *Propionibacterium* genera contributed to the higher *Actinobacteria* phylum abundance characteristic of healthy vessel walls (Figure 2).

Comparing the beta diversity of fecal, blood, thrombus and aneurysm wall microbiome compositions, separate groups of sample types are exhibited in the PCoA diagram, Figure 3. There was also a significant difference between the beta diversity of the aneurysm wall and thrombus samples (*p* = 0.014), but in all further groupings, a difference of *p* = 0.001 can be calculated between the cohorts. The blood samples are located between the feces and aneurysm-thrombus samples, in accordance with their assumed mediating role in the 3D diagram.

The occurrence of the most common 25 genera in the different sample types is shown in the heatmap, Figure 4. *Sphingobium*, *Acinetobacter*, *Burkholderia* and *Aquabacterium* genera, which have the highest abundance in aneurysms and thrombus, were present in blood and feces with very low abundance. The genera *Bacteroides*, *Prevotella*, *Finegoldia*, *Peptoniphilus* and *Anaerococcus*, which are dominantly present in the faecal microbiome, did not enter the bloodstream in detectable quantities. *Escherichia* and *Escherichia-Shigella* genera, which are present in high abundance in the blood, may originate from the gut microbiome, since these bacteria were also present there. Additional genera that occur in significant quantities in the blood microbiome, such as *Propionibacterium*, *Enhydrobacter* or *Moraxella*, cannot be derived from the stool microbiome.

The genera present in the different samples with significantly different abundances were compared using the LEfSe (Linear discriminant analysis effect size) biomarker discovery analysis (Figure 5). Comparing the different sample types to the stool samples, *Enhydrobacter* and *Cutibacterium* appeared as new genera in the blood, and these genera continued to have significantly higher abundance in the thrombus and aneurysm vessel wall. *Acinetobacter* and *Aquabacterium* were new genera with significantly higher abundances in the thrombus compared to the feces, and the *Sphingobium* genus also appeared newly in the aneurysm wall. Compared to the circulating blood, the genera *Sphingobium*, *Acinetobacter* and *Aquabacterium* were present in significantly higher quantities both in the thrombus and the aneurysm vessel wall, but higher amounts of the genera *Enhydrobacter* and *Cutibacterium* remained in the blood.

Functional capabilities and quantities of the microbial communities were predicted and profiled based on the observed amplicon content using the PICRUST2 tool of CosmosId [22]. The bacterial enzymatic activity that is likely present in a given sample was significantly different between the healthy control vascular wall and the aneurysm vessel wall samples (Figure 6). The set of enzymes characteristic of the microbiome associated with a healthy vascular wall includes significantly more Iron-chelate-transporting ATPase, Polar-amino-acid-transporting ATPase, Monosaccharide-transporting ATPase, N-acetylmuramoyl-L-alanine amidase and Non-specific serine/threonine protein kinase as the aneurysm microbiome enzyme set. The latter contains greater amounts of Glutathione transferase, NADH:ubiquinone reductase, Acetyl-CoA C-acetyltransferase, Peptidylprolyl isomerase and Enoyl-CoA hydratase both in terms of statistical significance and biological relevance.

## 3. Discussion

To the best of our knowledge, this is the first account of microbiome detection from the AAA vessel wall and associated thrombus. With the 16S rRNA detection method that we have successfully used before, we were able to perform a microbiome analysis with an appreciably high read count from all of our tested samples, whether from blood or vessel walls [21,23]. Nakayama and colleagues searched for the DNA of 11 bacterial species in the aneurysm wall and in the blood by RT-PCR method, but these 11 selected taxa could only be detected in the aneurysm of the 30 patients in 11 cases and in the blood in 19 cases. By verifying the presence of intestinal bacteria, they documented the translocation of bacteria from the gut through the blood to the vessel wall [17].

Studies attempting to detect the presence of odontopathogenic bacteria using different PCR techniques detected these bacteria in varying proportions (0–86%) in the aneurysm wall of AAA patients [18,19,20].

During our previous research, we showed that bacterial DNA can also be detected in the healthy vessel wall, which can come either from the oral cavity or from the gut microbiome, such as *Porphyromonas*, *Prevotella*, *Corynebacterium* or *Akkermansia*, *Escherichia*, *Enterobacter*, *Ruminococcus* genera [21]. In the current research, we investigated the origin of the bacterial taxa appearing in the AAA vessel wall, which is different from the healthy vessel wall. Compared to the taxon abundance at the phylum level of the healthy and AAA vessel wall, a shift is visible that was found to be typical of the faecal microbiome of AAA patients in a previous study [5], with an increased amount of *Proteobacteria* and a decrease in the abundance of *Bacteroidetes* and *Firmicutes*. The genera that cause the rise of *Proteobacteria* partially can also originate from the gut microbiome, such as for example *Escherichia* or *Escherichia-Shigella.* These bacteria can enter the thrombus and vessel wall through the blood after a preliminary gut dysbiosis and bacterial translocation for the patients we examined. The result obtained by Tian et al. with the dominance of *Proteobacteria* in the feces of the 21 patients we examined was confirmed in only seven cases. For other patients, the order of dominance supported the findings of Ito et al., *Firmicutes*, *Bacteroides*, *Proteobacteria* and *Actinobacteria* [4]. Among the bacteria present in the faecal microbiome, in human samples, the protective effect against AAA can be assumed for the species *Bifidobacterium adolescentis* [4] and *Roseburia intestinalis* [5], and in mouse models, the protective effect of *Akkermansia muciniphila* has been confirmed [10,13]. Among the above-mentioned protective symbiotic commensals of gut microbiota implicated in AAA, only *A. mucinophila* was found in a stool sample of one of our examined patients with quantities higher than 1%. *Campylobacter gracilis*, characterized as a pathobiont [24], was present in the stool samples of our patients with variable abundance (0.1–5.5%). In an animal experiment, a specific pathological subtype of *Faecalibacterium prauznitzii* was documented to help the pathomechanism of AAA [13], the strains belonging to this species were present in fecal samples of our patients at 0.5–6.2%.

Exploring the path of the bacteria to the thrombus or to the aneurysm wall, based on our LEfSe analysis, we observed that *Finegoldia*, *Peptoniphilus*, *Bacteroides*, *Anaerococcus* and *Prevotella* genera, which have a high abundance in the feces, did not translocate into the blood. The genera *Enhydrobacter* and *Cutibacterium* appearing in the blood cannot be derived from the feces, perhaps for these, the skin can be the entry point [25,26]. Ensuing, they are transferred from the blood to the thrombus and partially to the aneurysm wall. However, the blood sample taken on the day of surgery does not contain the typical bacteria of the thrombus or aneurysm wall, the genera *Sphingobium*, *Acinetobacter* and *Aquabacterium*. We speculate that these bacteria could have been present earlier in the process of aneurysm and thrombus formation, and did not come to the pathological sites from the same day’s blood circulation. *Sphingobium*, *Acinetobacter* and *Aquabacterium* are ubiquitously distributed in various environmental matrices, such as soil and water, and are common inhabitants of the intestinal tracts of animals [27,28]. Artificial products used by humans, are not necessarily characterized by human-associated microbiomes; instead, they have their own microbiome that was formed for possible extreme selection effects in the human environment [29]. Our everyday objects subjected to cleaning, including machines, coffee machines, dishwashers, saunas, refrigerators or air conditioning systems in their microbiome composition contain significant amounts of *Acinetobacter*, *Sphingobium*, *Aquabacterium* and *Enhydrobacter* [30]. It is not surprising that these bacteria have been detected in different parts of the human body (skin, oral cavity, ocular surface, duodenum, blood) and in different diseases [31,32,33,34]. Patients with rheumatoid arthritis (RA) have an elevated risk of developing aortic aneurysms compared to the general population [35]. Interestingly, our AAA-associated microbiome results coincide with the finding that the presence of *Sphingobium* and *Aquabacterium* in the saliva samples of patients with periodontitis and the amount of host inflammatory mediators can be considered predictive biomarkers associated with RA [36].

Apparently, in addition to the DNA of the bacteria identified by the 16S rRNA sequencing method, enzymes specific to bacteria are present or were present in the studied samples. The different enzyme activities characteristic of the healthy vessel wall and the aneurysm, which are effectors of various bacterial metabolic processes, can have a protective or enhancing effect on the formation of the aneurysm.

Local iron accumulation within the aortic wall plays a role in the pathogenesis of AAA through the promotion of oxidative stress and inflammation [37]. The complex process—as activated T lymphocites enhance macrophage iron accumulation, lipid peroxidation, and migration leading to the formation of AAA—could be inhibited in a mouse experiment using the iron chelating agent deferoxamine mesylate [38]. The high Iron-chelate transporting ATPase activity of bacteria in the healthy vessel wall can have an identical protective effect against the formation of AAA.

Metabolomic analysis of plasma from patients with AAAs has identified abnormalities in the metabolism of amino acids, including alanine, aspartate, glutamate, arginine, and proline. The precise relationship between these metabolites and AAAs is not yet clear, but a correlation was confirmed between the amount of the aforementioned amino acids and the size of the AAA [39]. Polar-amino-acid-transporting ATPase, the enzyme, found in bacteria of healthy vessel walls, may interact with an extracytoplasmic substrate binding protein and mediates the import of polar amino acids, providing protection against AAA progression [40].

In AAA patients the human Peptidyl-prolyl isomerase (PPI) expression is also enhanced in blood vessel tissue mostly in smooth muscle cells (SMCs), and murine experiments confirmed that Peptidyl-prolyl isomerase deficiency attenuates angiotensin II-induced abdominal aortic aneurysm formation [41]. The human PPI Pin1 appears to be a central regulator of SMC dysfunction and elastin degradation in the context of AAA formation. The high bacterial PPI enzyme activity of the aneurysm vessel wall may further worsen the pathogenesis of AAA.

## 4. Materials and Methods

### 4.1. Sample Collection

In our prospective study, 21 abdominal aortic aneurysm patients who underwent surgical repair between March 2022 and February 2023 at the Department of Vascular and Endovascular Surgery, Heart and Vascular Center, Semmelweis University were included. The inclusion criterion for the study was coverage of the aneurysm with thrombus. Patient No 22 was excluded from further comparative analysis of the study because the microbiome analysis of all samples showed a high abundance of *Mycobacterium tuberculosis*. Samples of the aneurysmal wall and intraluminal thrombus were carefully collected, to avoid contamination in the open aortic repair operation. About 3 g of the aneurysmal wall and intraluminal thrombus were collected. Blood was washed from these samples with sterile saline, then samples were snap-frozen with liquid Nitrogen. At least 3 mL of whole blood was collected into citrate-filled VACUETTE collection tubes (Greiner Bio-One, Stonehouse, UK), and after tube rotation, immediately frozen to –80 °C. Faecal samples were collected on the second day before the operation after admission. All the samples were stored at −80 °C until DNA extraction.

As a negative control for the blood vessel wall microbiome comparisons, we used the results of healthy vessel wall microbiome analysis from our previous research [21].

### 4.2. DNA Extraction

DNA isolation was performed by NucleoSpin Blood, Mini kit (Macherey-Nagel, Allentown, PA, USA) from blood samples and by ZymoBIOMICS DNA Miniprep Kit (Zymo Research Corp., Irvine, CA, USA) from aneurysm, thrombus and stool samples according to the manufacturer’s instructions, after enzymatic dissolution with ProtK (56 °C, 3 h).

### 4.3. 16S rRNA Microbiota Analysis

Bacterial DNA was amplified with tagged primers covering the V3-V4 region of the bacterial 16S rRNA gene. PCR and DNA purifications were performed according to Illumina’s protocol. PCR product libraries were assessed using a DNA 7500 Kit with Agilent 2100 Bioanalyzer (Agilent Technologies, Waldbronn, Germany). Equimolar concentrations of libraries were pooled and sequenced on Illumina MiSeq platform (Illumina, San Diego, CA, USA) using MiSeq Reagent Kit v3 (600 cycles PE) Illumina, Inc. (Berlin, Germany). To avoid contamination and to increase the reliability of the study, all analysis procedures were performed in duplicate. In order to evaluate the contribution of extraneous DNA from reagents, extraction negative controls and PCR negative controls as well as PCR positive controls (ZymoBIOMICS Microbial Community Standard, Zymo Research Corp., Irvine, CA, USA) were included in every run. Raw sequencing data were retrieved from the Illumina BaseSpace and the data were analysed by the CosmosID bioinformatics platform [22].

### 4.4. Statistical Analysis

The Wilcoxon Rank Sum test for Chao1 Alpha diversity, and PERMANOVA analysis for Bray–Curtis PCoA Beta diversity were used for statistical testing between cohorts of samples, by applying the statistical analysis support of CosmosID bioinformatics platform. Statistical significance was decided upon at a two-tailed *p* value of ≤0.05. Also, the CosmosId bioinformatics platform was used for LEfSe (Linear discriminant analysis Effect Size) to identify features including taxa, and enzymatic pathways characterizing the differences between two cohorts (Cosmosid Inc., Germantown, MD, USA).

## 5. Conclusions

In conclusion, this study explores the microbiome in the aneurysm wall, thrombus, blood and stool of AAA patients. A significant difference was confirmed between the microbiome composition of the aneurysm wall and the healthy vessel wall. Aneurysm-associated bacteria can only be partially derived from the dysbiotic gut microbiome. *Acinetobacter*, *Aquabacterium* and *Sphingobium* species, which are most characteristic of aneurysms, could have reached the vessel wall from the environment either through the skin or the oral cavity. Bacterial enzyme activity characteristic of a healthy vessel wall can inhibit, and that characteristic of an aneurysm can increase the progression of AAA.

## Figures and Tables

**Figure 1 ijms-25-08844-f001:**
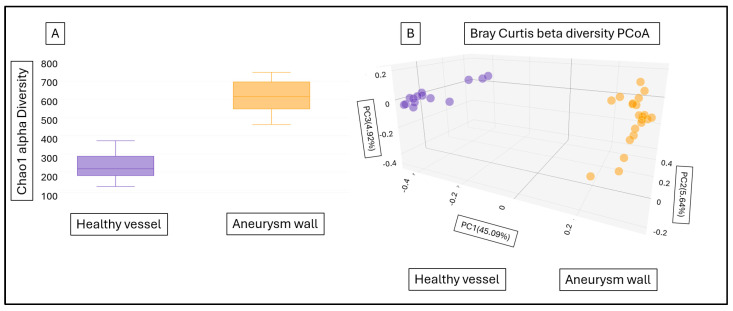
Chao1 alpha diversity (**A**) and Bray Curtis beta diversity PCoA (**B**) of healthy vessel and aneurysm wall microbiome. Purple: healthy vessel microbiome, yellow: aneurysm wall microbiome.

**Figure 2 ijms-25-08844-f002:**
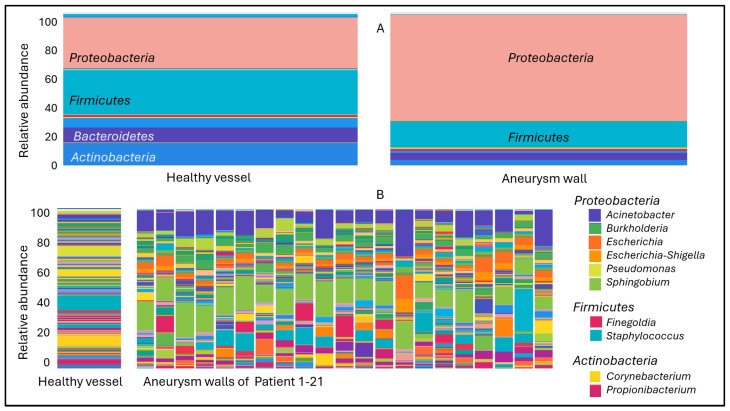
Relative taxon abundance of healthy and aneurysm vessel wall microbiome composition, (**A**) at phylum level, (**B**) at genus level.

**Figure 3 ijms-25-08844-f003:**
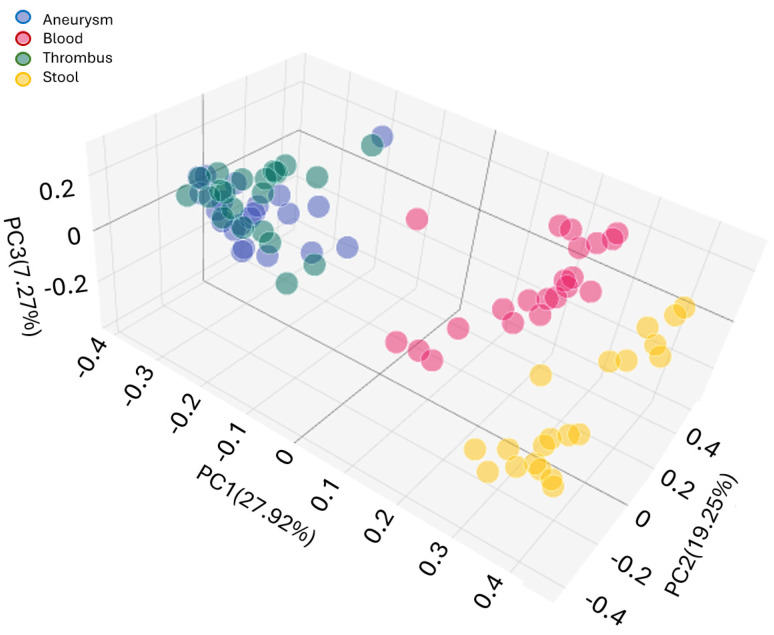
Bray Curtis beta diversity PCoA of stool (yellow), blood (red), thrombus (green) and aneurysm (blue) microbiome of AAA patients.

**Figure 4 ijms-25-08844-f004:**
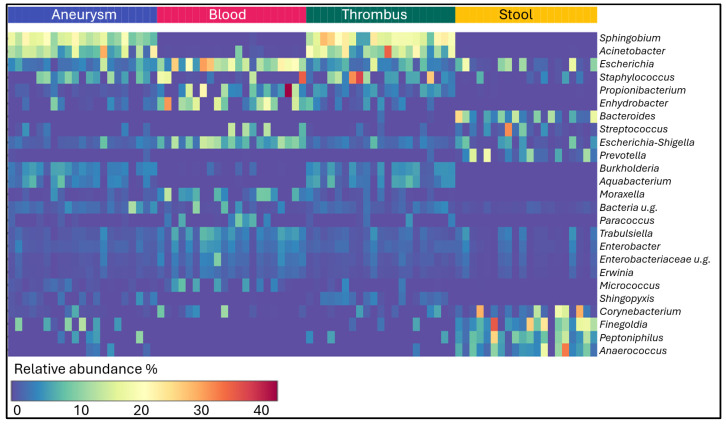
Heatmap about the most abundant 25 genera in aneurysm, blood, thrombus and stool samples.

**Figure 5 ijms-25-08844-f005:**
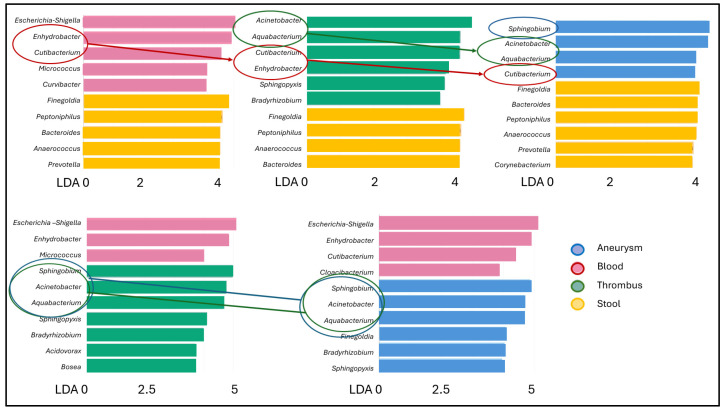
LEfSe bar chart visual representation of discriminative feature of genera abundances among the different sample types. LDA is the Linear Discriminant Analysis score.

**Figure 6 ijms-25-08844-f006:**
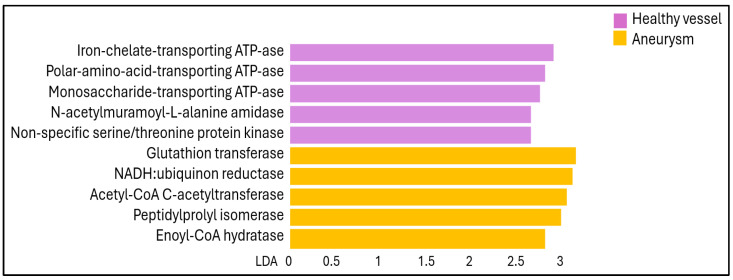
LEfSe bar chart representation of significantly different predicted enzyme activity among the microbiome compositions of healthy control and aneurysm vessel walls.

## Data Availability

The datasets generated and analyzed during the current study are available in the SRA repository: SRA/PRJNA1128157/www.ncbi.nlm.nih.gov (accessed on 26 June 2024).

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
