# Peer review of "Relationship between Gut, Blood, Aneurysm Wall and Thrombus Microbiome in Abdominal Aortic Aneurysm Patients"

_ijms, 2024, doi:10.3390/ijms25168844_

Round 1
Reviewer 1 Report
Comments and Suggestions for Authors
The manuscript entitled Relationship between gut, blood, aneurysm wall and thrombus microbiome in Abdominal Aortic Aneurysm Patients by Éva Nemes-Nikodém et al in which the authors studied the stool, 20 blood, thrombus and aneurysm microbiomes of abdominal aortic aneurysm patients using 16S rRNA sequencing to get insights into the pathways bacteria translocate 22 into the aneurysm.
The manuscript needs extensive revisions before it gets accepted.
The authors must revise their manuscript according to the following comments.
Abstract
Should be restructured as aim of the study / research work is not clear.
Introduction
How Abdominal aortic aneurysm (AAA) is linked with Gut microbiota?
Line 67-69
DNA of selected bacteria originating from the intestinal or oral microbiome is positively detectable in the vascular, thrombus or blood samples of AAA patients, suggesting that bacteria can participate in the progression of the aneurysm with their direct presence [16-19].
How the Microbiome is influenced by direct presence? What does this sentence mean?
What is the role of Diet with AAA and Gut Microbiota and other inflammatory diseases.
The authors must revise it.
In-fact the whole introduction should be re-written.
The following articles could be of good help.
1) Elucidating the Role of Diet in maintaining the Gut Health to Reduce the Risk of Obesity, Cardiovascular and other age-related Inflammatory Diseases: Recent Challenges and Future Recommendations.
2) Dietary Implications of the Bidirectional Relationship between the Gut Microflora and Inflammatory Diseases with special emphasis on Irritable Bowel Disease: Current and Future Perspective.
3) Lacticaseibacillus paracasei BNCC345679 Revolutionises DSS Induced-Colitis and Modulates Gut Microbiota.
If the authors wants to cite these article its their choice.
The results are well explained.
Discussion is fine.
The conclusion should be revised.
The methodology is well designed.
Figures quality is good.
Reviewer 2 Report
Comments and Suggestions for Authors
The manuscript "The relationship between intestine, blood, aneurysm wall and thrombus microbiome 2 in patients with abdominal aortic aneurysm" is a very interesting and up-to-date article. The well-written manuscript, correctly collected data, coherently presented results, however, require small corrections.
In figure 5 and figure 6, inside the LEfSe bar chart, the writing is indecipherable. Please improve the resolution. And to the statistical analysis, please add the name, model, and country of the biostatistics software used.
Overall, it is an interesting paper with important information about the behavior of abdominal aortic aneurysm.
Reviewer 3 Report
Comments and Suggestions for Authors
The manuscript is well written. However, some aspects should be addressed before considering for publication.
#1. Given the descriptive nature of the study, any speculation reported in the abstract or in the text should be avoided. As an example, authors cannot argue the time in which bacteria arrived within thrombus or arterial wall, yet they should better described potential differences with those found into blood flow.
#2. On page 2, lines 77-83, three main goals are presented here, but not mentioned in the abstract nor commented in the conclusion. My suggestion is to consider one or two main purpose of the analysis and modify the above mentioned sections accordingly.
#3. Part B of figure 1 is unclear and should be better represented as well as part A. Similarly, circles and lines added upon figure 5 should be removed.
#4. Introduction is too long and must be shortened. There are no sections Results nor Discussion.
Round 2
Reviewer 1 Report
Comments and Suggestions for Authors
The authors have revised the manuscript and it can be accepted for publication now.
Author Response
Dear Reviewer 1,
We would once again like to thank You for having such a positive attitude towards our paper.
Sincerelly yours
Eszter Ostorházi
Reviewer 3 Report
Comments and Suggestions for Authors
Authors did not change the structure of the manuscript. Introduction is too long. There is no Discussion. The manuscript ends with statistical analysis, which is unusual. The manuscript cannot be accepted for publication in its present form.
Author Response
Dear Reviewer 3,
We would like to respond to your 2nd round of review, we have done everything possible to improve the manuscript in accordance with your request. Your review was as follows:
"Authors did not change the structure of the manuscript. Introduction is too long. There is no Discussion. The manuscript ends with statistical analysis, which is unusual. The manuscript cannot be accepted for publication in its present form. "
IJMS prescribes a fixed format for the preparation of manuscripts, this structure is not allowed to be changed. IJMS Microsoft Word template file
We have reformulated and shortened the Introduction, now it is only 37 lines long instead of 46.
The Discussion chapter is located before the Materials and Methods chapter in accordance with the IJMS regulations, it is not missing from the article. We highlighted the Discussion in yellow, which can be found between lines 205-296.
The description of Statistical analysis methods are part of the Materials and Methods section. An important part of the research article on microbiome research is the precise description of all methodologies, including the statistical methods used. We will not delete this section from the manuscript.
The manuscript ends with the Conclusion chapter.
We hope that you now find the Introduction chapter short enough. We trust that, knowing the use of the IJMS mandatory format, you will no longer blame the structure of the manuscript as an error. We hope that you find the manuscript suitable for publication in its current form.
Best regards
Eszter Ostorhazi
Round 3
Reviewer 3 Report
Comments and Suggestions for Authors
My opinion about the article remains the same. No further comments.